# Evaluation of Temozolomide and Fingolimod Treatments in Glioblastoma Preclinical Models

**DOI:** 10.3390/cancers15184478

**Published:** 2023-09-08

**Authors:** Mélodie Davy, Laurie Genest, Christophe Legrand, Océane Pelouin, Guillaume Froget, Vincent Castagné, Tristan Rupp

**Affiliations:** Porsolt SAS, ZA de Glatigné, 53940 Le Genest-Saint-Isle, France

**Keywords:** brain cancer, glioblastoma, glioma, preclinical models, tumor progression, Fingolimod, Temozolomide

## Abstract

**Simple Summary:**

Grade 4 gliomas and glioblastomas are the most common brain tumors, accounting for 50% of primary brain tumors, and the most aggressive, with a median 5-year patient survival of less than 10%. There is no therapy to date that can sustainably prolong the life of patients. In this work, we demonstrated that the standard of care, Temozolomide, and Fingolimod, an immunomodulatory molecule with expected anti-cancer activity, reduced tumor cell survival in vitro. Conversely, Temozolomide reduced tumor growth in vivo in mouse and human orthotopic glioblastoma models, while Fingolimod did not. Globally, our data suggest that the Temozolomide response varies depending on the type of cancer model and that the efficacy of Fingolimod may still need confirmation.

**Abstract:**

Glioblastomas are malignant brain tumors which remain lethal due to their aggressive and invasive nature. The standard treatment combines surgical resection, radiotherapy, and chemotherapy using Temozolomide, albeit with a minor impact on patient prognosis (15 months median survival). New therapies evaluated in preclinical translational models are therefore still required to improve patient survival and quality of life. In this preclinical study, we evaluated the effect of Temozolomide in different models of glioblastoma. We also aimed to investigate the efficacy of Fingolimod, an immunomodulatory drug for multiple sclerosis also described as an inhibitor of the sphingosine-1-phosphate (S1P)/S1P receptor axis. The effects of Fingolimod and Temozolomide were analyzed with in vitro 2D and 3D cellular assay and in vivo models using mouse and human glioblastoma cells implanted in immunocompetent or immunodeficient mice, respectively. We demonstrated both in in vitro and in vivo models that Temozolomide has a varied effect depending on the tumor type (i.e., U87MG, U118MG, U138MG, and GL261), demonstrating sensitivity, acquired resistance, and purely resistant tumor phenotypes, as observed in patients. Conversely, Fingolimod only reduced in vitro 2D tumor cell growth and increased cytotoxicity. Indeed, Fingolimod had little or no effect on 3D spheroid cytotoxicity and was devoid of effect on in vivo tumor progression in Temozolomide-sensitive models. These results suggest that the efficacy of Fingolimod is dependent on the glioblastoma tumor microenvironment. Globally, our data suggest that the response to Temozolomide varies depending on the cancer model, consistent with its clinical activity, whereas the potential activity of Fingolimod may merit further evaluation.

## 1. Introduction

Gliomas and glioblastomas (GBMs) are a critical public health concern affecting patient survival and quality of life. Advanced glioma and in particular glioblastoma is quite rare, with a low incidence of 3.2 to 3.3 cases per 100,000 patients in the United States and France, respectively [1,2]. GBM remains difficult to treat due to its per se localization and aggressiveness, making it the deadliest cancer [3]. Despite numerous clinical trials, the standard treatment remains surgical resection and the Stupp protocol which combines radiotherapy and Temozolomide (TMZ) as chemotherapy [4,5]. However, this strategy improves median survival by approximately 3 months, i.e., 15 months versus 12 months for surgery and radiotherapy only [4]. A growing body of work shows that molecular heterogeneity between and within tumors likely contributes to the low clinical benefit observed [6,7,8]. Characterizing this heterogeneity and developing new predictive models to assess the efficacy of innovative therapeutic strategies remain both preclinical and clinical challenges.

GBMs express the native form of the isocitrate dehydrogenase gene (IDH1) and nuclear retention of alpha-thalassemia/intellectual disability, X-linked (ATRX) [9]. In addition, GBMs are characterized by a high rate of mitotic cells, extensive necrosis, nuclear pleomorphism and rich vascularization, which contributes to their aggressiveness and morbidity [10]. Epigenetic disturbances such as the methylation of the O(6)-methylguanine-DNA methyltransferase (MGMT) promoter are observed in GBM patients, which is associated with a better prognosis and chemotherapy sensitivity [11,12]. Nevertheless, all GBMs exhibit resistance to chemotherapy or targeted therapy leading to very poor survival [13]. This multiplicity of etiological factors complicates clinical treatment and the preclinical evaluation of new drugs. Therefore, the development of relevant in vitro and in vivo preclinical models to test the efficacy and safety of new therapeutic strategies is still required.

In vitro models allow a reductionist approach to dissect the cellular processes involved in tumorigenesis. However, performing 2D standard cell culture experiments does not provide a cell environment that mimics the in vivo situation, while 3D cell culture provide more accurate cell-to-cell interactions or tumor architecture [14]. Despite their advantages, in vitro 3D cell models only offer a limited field of analysis compared to the complexity of the in vivo tumor microenvironment. Indeed, animal models allow for a more complete analysis of the interaction between different cell types, whether tumor cells or stroma [15]. Human and mouse cells are commonly used for the in vivo evaluation of test substances. Whereas the use of human cells requires immunodeficient mice lacking a fully functional immune system, the syngeneic models provide a complete immune response, allowing the analysis of the role of immune system in the drug response. Syngeneic and xenograft models thus display complementary aspects of tumor progression that should be evaluated [16,17]. It remains important to characterize preclinical models to better predict the occurrence and development of GBM, and to test new therapeutic options likely to improve the quality of life of patients compared to available treatments.

TMZ is described as inducing hepatic, hematological, and gastrointestinal toxicity in glioma patients [4,18]. However, in contrast to chemotherapies, which can induces cytotoxicity in healthy and tumor cells, targeted therapies tackle specific mechanisms upregulated in the tumor microenvironment (TME) or involved in cancer, limiting the effect in healthy tissues [19]. Interestingly, new innovative therapeutics for GBM are under evaluation, including lipid-based molecules [20]. Recently, growing evidence has demonstrated the ability of sphingosine-1-phosphate (S1P), as a lipid-related molecule, to contribute to cancer and potentially to GBM [21,22]. S1P acts as an agonist and activates five described S1P receptors (S1PR) contributing to cell survival, migration, angiogenesis, and immune cell trafficking [23]. Different inhibitors of the S1P/S1PRs axis have been developed, including Fingolimod, which is approved for the treatment of multiple sclerosis. Fingolimod retains T cells in secondary lymph organs through the inhibition of S1PRs, including S1PR1, S1PR3, S1PR4, and S1PR5 [24,25], limiting T cell brain infiltration and lowering disease progression [26]. Interestingly, preclinical studies suggest that Fingolimod may favor an anti-tumoral response for different types of cancer, such as triple-negative breast cancer and lung cancer [25,27,28,29,30]. Fingolimod can cross the blood–brain barrier as radiolabeled fingolimod accumulates in the white matter of the central nervous system (CNS) [31]. Fingolimod affects cells in the CNS and might even induce neuro-regeneration [32].

Interestingly, in vitro Fingolimod reduced the viability of glioma cells without affecting normal cells, such as primary astrocytes [33,34,35]. Fingolimod also diminished the migration and invasion of GBM cells [34,36,37]. Different mechanisms of action have been described to affect GBM cells, including mTOR/AKT signaling, STAT3 signaling, NRF2 signaling, p53 expression, and metalloprotease expression [33,34,35,37,38,39]. Some data suggest that Fingolimod displayed anti-tumoral effects in vivo in rat orthotopic glioma models and also when using human GBM cells injected subcutaneously, even if no effect was observed on animal survival [35,36]. Further preclinical evaluation remains important to determine the potential utility of Fingolimod in GBM.

Here, we investigated the effect of TMZ and Fingolimod treatments in GBM in vitro and in vivo models. We used different GBM cell lines, i.e., IDH1wt and MGMT hypermethylated U87MG human cells [40,41,42,43,44], IDH1wt and low MGMT methylated U118MG human cells [43,45,46], IDH1wt and unmethylated MGMT U138MG human cells [43,44], and IDH1wt and MGMT hypermethylated GL261 mouse cells recapitulating human GBM features [16,47,48,49,50]. Using in vitro 2D and 3D culture models, we demonstrated that TMZ displayed cytotoxic effects on the four GBM cells tested. Moreover, we demonstrated that TMZ had a varied effect on different cellular models, with a strong anti-tumor effect on U87MG and GL261 tumors, while U118MG and U138MG tumors were less affected. Independently, we observed through clinical survival analysis that S1PR2 and S1PR3 are overexpressed in GBM and low-grade glioma (LGG) tumors. Moreover, a high expression of S1PR4 for GBM patients and S1PR3 for LGG patients is correlated with poor prognosis, suggesting that targeting S1PR through an inhibitor like Fingolimod might be relevant in glioma. In vitro, Fingolimod induced cytotoxicity in 2D culture but had little or no effect in 3D spheroid culture. Moreover, Fingolimod alone or in combination with TMZ did not affect GBM in vivo growth alone or in combination with TMZ in GL261 allograft and U87MG xenograft orthotopic GBM models. To our knowledge, this study evaluates for the first time the effect of Fingolimod in well-described GL261 and U87MG GBM models without observing anti-tumor effects. This suggests that further work is required in order to evaluate the interest of Fingolimod as an anti-cancer therapy in GBM.

## 2. Material and Methods

### 2.1. Cells and Cell Culture

The cell culture procedure was carried out according to previous work [28,51]. U118MG human glioblastoma cells (HTB-15™ obtained from ATCC^®^, Manassas, VA, USA), GL261 mouse glioma cells (ACC 802 obtained from Leibniz Institute DMSZ), U138MG human glioblastoma cells (HTB-16™ obtained from ATCC^®^, Manassas, VA, USA), and U87MG human glioblastoma cells (ECACC 89081402™ obtained from ATCC^®^, Manassas, VA, USA) were cultured with RPMI 1640 (Gibco^®^, Billings, MT, USA, ATCC-formulated) supplemented with fetal bovine serum (FBS, Gibco^®^) at a final concentration of 10% including antibiotics (Penicillin 100 U/mL-Streptomycin 100 µg/mL, Gibco^®^). The U87MG-RFP-Luc cells (transduced with lentiviral biscitronic vector LVP009-PBS from AMSBIO, Abingdon, UK) expressing firefly luciferase (Luc) and red fluorescence protein (RFP) were cultured in RPMI medium as described above, including 8 μg/mL of Puromycin. The GL261-Luc cells (transduced with lentiviral vector 79692-G from AMSBIO) expressing luciferase (Luc) were cultured in RPMI medium as described above, including 100 μg/mL of G418/Geneticin. All procedures were performed under aseptic conditions as previously described [28,51]. Cells were grown at 37 °C and 5% CO_2_ and they did not expand beyond 8 passages in culture. All cell lines tested negative for mycoplasma just prior to the experiment sessions using the MycoAlert^®^ Mycoplasma Detection Kit (reference LT07-318, Lonza™, Basel, Switzerland). Before cell injection, cells reaching 70–90% confluence were split and the cell count was assessed using the cell counter Nucleocounter NC-200™ (ChemoMetec^®^, Allerod, Denmark).

### 2.2. In Vitro Treatments

Cells were treated in 96-well plates. TMZ was used at 0.01, 0.1, 1, 10, 50, 100, 250, 500, and 1000 µM [45] and was diluted in culture medium containing up to 0.5% dimethylsulfoxide (DMSO, reference D8418, Merck^®^, Rahway, NJ, USA). Fingolimod was used at 0.1, 0.5, 1, 2.5, 5, and 10 µM [29,33] and was diluted in culture medium; above these doses Fingolimod was not fully soluble.

### 2.3. 2D Cell Growth Assay

Cells were plated in 96-well plates (2000 cells/well with 3 to 4 replicates) for 24 h, before being treated with TMZ or Fingolimod. In image-based cytotoxicity analysis, cells were cultured for an additional 72 h in culture medium containing fluorescent DNA intercalating agent, which was used to detect dead cells that were stained when their plasma membrane was compromised (Sytox™ green/red Dead Cell Stain, reference S7020, ThermoFisher Scientific™, Waltham, MA, USA). Then, cell cytotoxicity was measured as a positive fluorescent object and was counted via image-based analysis using the Ensight™ system (Perkin Elmer^®^, Waltham, MA, USA).

Alternatively, in the cell growth assay, cells were cultured in culture medium and treated with Fingolimod. After 72 h of growth, MTS (3-(4,5-dimethylthiazol-2-yl)−5-(3carboxymethoxyphenyl)−2-(4-sulfophenyl)−2H-tetrazolium, inner salt) incorporation assays were performed according to the manufacturer’s recommendation (CellTiter 96^®^ Aqueous One Solution Cell Proliferation Assay, reference G358x, Promega^®^, Madison, WI, USA). Measured absorbance values at 490 nm were subtracted with absorbance at 690 nm (optical background) and were normalized to the control vehicle-treated conditions as relative cell growth values using a microplate reader, Ensight™ (Perkin Elmer^®^, Waltham, MA, USA). Three independent experiments were performed. All experiments were performed with at least three technical replicates.

### 2.4. 3D Tumor Spheroid Assay

The procedure was adapted from previous work [28]. Cells (i.e., U87MG, U118MG, U138MG, and GL261) were plated in 96-well CellCarrier Spheroid ULA Microplates™ (Perkin Elmer^®^) at 2000 cells/well in culture medium, including 3–4 replicates per experimental condition. Spontaneously formed tumor spheroids appeared after 48 to 74 h of growth. Then, spheroids were treated with TMZ or Fingolimod in culture medium containing fluorescent DNA intercalating agent, which detects dead cells that are stained when their plasma membrane is compromised (Sytox™ green Dead Cell Stain, reference S7020, ThermoFisher Scientific™). Confluence and the positive fluorescence area were monitored by image-based analysis using the Incucyte™ system (Sartorius^®^, Goettingen, Germany). One image every 4 h was acquired. Spheroid growth was determined by analyzing the total spheroid surface (in pixel^2^). All data were normalized to the control condition per experiment. Cytotoxicity was determined as the fluorescent positive area in pixel^2^ among the spheroid area. Cytotoxicity was defined as a percentage (%). Two to three independent experiments were performed per test substance. All experiments were performed with 3 to 4 technical replicates.

### 2.5. Animals

Housing was adapted according to previous work [29]. Female C57BL/6JRj or BALB/cAnN-Foxn1nu/nu/Rj (=BALB/c-nude) mice were supplied by Janvier Labs. Animals were acclimated at least 5 days before the implantation of tumor cells, which was performed on 7- or 8-week-old mice. C57BL/6JRj mice were housed with up to 10 animals per cage in a biosafety level 1 laboratory. BALB/c-nude mice were housed with up to 6 animals per cage in a biosafety level 2 laboratory in individually ventilated cages (NEXGEN MOUSE IVC™, Allentown^®^, Allentown, NJ, USA) with NestPak^®^ (Allentown^®^ Allentown, NJ, USA). Nesting enrichment was provided (tube, cotton, and wood). The room was maintained under artificial lighting (12 h) between 7 a.m. to 19 p.m. at 22 ± 2 °C. Mice received rodent diet (SAFE^®^ A04 or R04-40) and water ad libitum. Mice were identified by permanent tattoos. The number of mice per group is included in the figure legends for all the experimental designs.

### 2.6. In Vivo Treatments

TMZ and Fingolimod were diluted in 5% DMSO + 30% polyethylene glycol (PEG) 300 + H_2_O. The mice were randomized based on their tumor size and body weight once the tumors reached an approximate average volume of 100 mm^3^ for the subcutaneous model or at least 1 × 10^6^ photon/second/steradian/cm^2^ (p/s/sr/cm^2^) for orthotopic models. TMZ is used in the clinic for the treatment of human GBM at a dose of between 75 and 200 mg/m^2^/day for 5 days of every 28-day cycle [4,52]. The equivalent dose calculated for mice based on the Km value is between 25 and 66 mg/kg [53]. We observed that doses higher than 10 mg/kg induced toxicity in mice (internal data). Thus, mice were treated with TMZ at 1 and 10 mg/kg via the oral route in order to avoid co-morbidity induced by TMZ. Mice were treated with Fingolimod at 5 mg/kg via the oral route according to previous work [54,55]. Drugs or vehicles were administered five times per week until the end of the experiment.

### 2.7. Subcutaneous Graft Animal Model

The procedure for subcutaneous graft was adapted from [51]. For U118MG, U138MG, and U87MG, prepared cells resuspended in sterile phosphate-buffered saline (PBS) kept on ice were injected with 5 × 10^6^ cells subcutaneously into the right flank of the BALB/c-nude mice. GL261 cells were injected with 5 × 10^6^ cells subcutaneously into the right flank of C57BL/6JRj mice. Mice were anesthetized with 2% isoflurane (Axience^®^, reference 152678) at 2 L/min and were kept on a warming pad, and eye lubricant was applied during the procedure. The area of injection was shaved, for C57BL/6JRj only, and cleaned with Chlorhexidine (Antisept™, reference ANT015) before the injection of 100 µL of cell suspension (in an insulin syringe). The mice were monitored (breathing) until they woke up. Tumor volume was calculated using the formula V = (a^2^ × b)/2, where b is the longest axis and a is the perpendicular axis to b, and was measured two to three times per week with a caliper. Several physiological and behavioral parameters were monitored during the study including, if needed, temperature, dyspnea, eating and drinking, loss of balance, and sedation.

### 2.8. Brain Orthotopic Graft Animal Model

The procedure for orthotopic graft is adapted from [56]. GL261-Luc and U87MG-RFP-Luc cells were injected via stereotaxic apparatus with 1 × 10^5^ cells into the striatum of C57BL/6JRj or BALB/c-nude mice, respectively. Prior to the surgical procedure, mice were treated for analgesia with the nonsteroidal anti-inflammatory drug Carprofen at 5 mg/kg through the subcutaneous route.

The mice were anesthetized using isoflurane inhalation. Eye lubricant was applied during the procedure. The mice were placed on the bed of a stereotaxic apparatus and cells were injected into the striatum region. After exposing the top of the skull to visualize the Bregma (junction between the coronal and sagittal suture), a small hole was created in the skull at the position of 2 mm right (2 mm medio/lateral) and −1 or 1 mm anterior/posterior of the Bregma (according to Paxinos and Franklin, *The Mouse Brain*, second edition 2004). A 5–10 μL glass Hamilton syringe pre-loaded with cells was placed on a micro-pump system (Harvard Apparatus^®^, Holliston, MA, USA) and the tip of the needle was slowly advanced over a period of 4 min until it reached a depth of 3 mm (−3 mm ventro/dorsal). A defined volume of 4 µL of cells was infused at a rate of 0.5 μL per minute to limit any backflow. The needle was left in place for another 2 min, to allow all the cells to settle. Then, the needle was removed over a period of 4 min. The skull was closed with sterile bone wax. Then, around 100 µL of Lidocaine was applied onto the skull to limit post-surgical associated pain and the incision was sutured. Mice received Buprenorphine at 0.1 mg/kg through the subcutaneous route for complementary analgesia. The anesthetized mice were placed on a warming blanket and monitored (breathing) until they woke up.

As described in our previous work [28], tumor growth was monitored using in vivo bioluminescence imaging. During the procedure, animals were anesthetized and injected with D-luciferin (VivoGlo™, reference P1043) at 150 mg/mL through the intraperitoneal (i.p.) route. After 10 min, mice were imaged using an IVIS™ Lumina X5™ system (Perkin Elmer^®^) maintained on a warm blanket. Measurements were acquired twice a week to track the dynamics of tumor growth by measuring the luciferase activity expressed by the tumor cells. To quantify the bioluminescence signal, size-matched regions of interest (ROI) were obtained via an automated method using Living Image™ software (version 4.7.1., Perkin Elmer^®^), and signals were quantified as average bioluminescence (p/s/sr/cm^2^).

The Neurological Score was assessed according to a modified version of the method of Bederson et al. [57]. Spontaneous walking and circling toward the paretic side were first observed. Then, mice were placed on a horizontal bar by the forepaws to assess animal force and coordination. Then, the mice were held by the tail, placed on a rough surface, and their vibrissae/muzzle stimulated gently toward the ipsi- and contralateral sides to assess their lateral sensibility. Finally, mice were hung by the tail, with the right and the left hand of the experimenter, sequentially, and lifted above the bench to assess their body rotation. Each subtest was graded on a scale from 0 to 1 (0 = no response or totally abnormal response; 1 = normal response) or 0 to 2 (0 = no response or totally abnormal response; 1 = weak or abnormal response; 2 = normal response). A normal score fell between 9 and 11.

### 2.9. Animal Ethical Consideration and Limit Points

The procedure was adapted from previous work and an internal procedure [56]. All methods were designed to minimize animal suffering. Experiments were conducted in strict accordance with Council Directive No. 2010/63/UE of 22 September 2010 and the French decree No. 2013-118 of 1 February 2013 on the protection of animals for use and care of laboratory animals. The study was performed in an accredited lab from the Association for Assessment and Accreditation of Laboratory Animal Care (AAALAC). All experiments were also approved by the ethics committee for animal experimentation (agreement n° F 53 1031). The following parameters were considered as limit points that required mice sacrifice by CO_2_ inhalation or cervical dislocation: tumor volume exceeding 2000 mm^3^ (when injected subcutaneously), a body weight loss greater than 20% relative to the initial weight for two consecutive measures, high tumor necrosis or ulceration, hypothermia (<34 °C), dyspnea, failure to eat and drink, loss of balance, and marked sedation.

### 2.10. Patient Data Analysis

The clinical data from cohorts shown here are based on mixed publicly available data from the following sources: The Cancer Genome Atlas (TCGA) and The Genotype-Tissue Expression (GTEx) project using GBM and LGG cohorts [58,59]. Overall survival (OS) was analyzed by the Kaplan–Meier method with the online software ‘Kaplan-Meier Plotter’ from http://gepia.cancer-pku.cn/ [60]. Patients were stratified into low- and high-expression populations using the median as the cut off. The log-rank test was used for drawing comparisons between the low- and high-expression groups. Moreover, the expression level of the different genes was also analyzed with the online software ‘Expression DIY-Boxplot’ from http://gepia.cancer-pku.cn/ (accessed on 21 November 2022) [60]. The statistical analysis was carried out via a one-way ANOVA, using disease state (Tumor or Normal) as a variable for calculating differential expression. The 204642_at (S1PR1), 227684_at (S1PR2), 228176_at (S1PR3), 206437_at (S1PR4), and 230464_at (S1PR5) Affymetrix probe sets were used. 

### 2.11. Statistics

The procedures were adapted from previous work [29]. Statistical analyses and graphical representations were performed using GraphPad Prism (version 9.4.1). *p* values < 0.05 were considered as statistically significant (* *p* < 0.05; ** *p* < 0.01; *** *p* < 0.001; **** *p* < 0.0001). Data were tested for normality using the D’Agostino-Pearson test.

Tumor volume, body weight, and neuroscore data were analyzed for each day using a mixed-effects model (REML) (group and day as factors) with repeated measures alongside Tukey’s multiple comparison tests (for each day).

For in vitro 3D spheroid growth and cytotoxicity, data were analyzed for each day using a mixed-effects model (REML) (group and day as factors) with repeated measures alongside a Bonferroni’s multiple comparison test (versus control, for each day).

For in vitro 2D cell cytotoxicity, data were analyzed using a Kruskal-Wallis test with Dunn’s multiple comparison tests.

The cumulative survival was evaluated for significance with the log rank test.

## 3. Results

### 3.1. TMZ Induced GBM Cytotoxicity in Both 2D and 3D In Vitro Models

In order to analyze the effect of TMZ on tumor cell viability, we evaluated the TMZ dose–response relationship in GBM cell lines in 2D and 3D settings. In 2D culture setting, we demonstrated that TMZ induced a dose-proportional increase in cytotoxicity, which was significant for U87MG cells starting from the dose of 250 µM (Appendix A) and for U118MG from the dose of 500 µM (Appendix A). We also used a 3D culture system using GBM spheroid models. The 3D tumor spheroids mimic some aspects of the cell–cell interaction, hypoxia, or molecular heterogeneity of tumors [61,62], which are not reproduced by 2D cell culture [63], making spheroids more relevant models. First, we demonstrated that U87MG, U118MG, U138MG, and GL261 cells generated cohesive tumor spheroids within 48 to 120 h of culture in ultra-low-attachment round-bottom plates (Appendix A). Using this method, we demonstrated that TMZ significantly reduced the U87MG spheroid growth in a dose-dependent manner starting from 65 h at 100 µM and from 47 h at 1000 µM (Figure 1A). Moreover, TMZ induced a significant increase in cytotoxicity using U87MG spheroids starting from 69 h at 100 µM and from 41 h at 1000 µM (Figure 1B,C). Similarly, we demonstrated that TMZ significantly reduced the GL261 spheroid growth in a dose-dependent manner starting from 69 h at 10 µM, from 65 h at 100 µM, and from 37 h at 1000 µM (Appendix A). TMZ also induced a significant increase in cytotoxicity in GL261 spheroids starting from 65 h at 100 µM and from 33 h at 1000 µM (Appendix A). Furthermore, we also demonstrated that TMZ significantly reduced the U118MG spheroid growth starting from 29 h at 1000 µM (Figure 1D). TMZ induced a significant increase in cytotoxicity using U118MG spheroids starting from 33 h but only at 1000 µM (Figure 1E,F). Finally, we demonstrated that TMZ significantly reduced the U138MG spheroid growth in a dose-dependent manner starting from 57 h at 100 µM and from 37 h at 1000 µM (Appendix A). Likewise, TMZ induced a significant increase in cytotoxicity in U138MG spheroids starting from 53 h at 100 µM and from 25 h at 1000 µM (Appendix A). These data suggest that all GBM cell lines are sensitive to TMZ in vitro with an important anti-tumor response observed in U87MG and GL261 cell lines.

### 3.2. TMZ Affected GBM In Vivo Tumor Growth Differently in Subcutaneous Model

TMZ was evaluated in vivo in subcutaneous GL261 allograft and U87MG, U118MG, and U138MG xenograft models in single or repeated experiments. We demonstrated that TMZ at 10 mg/kg affected tumor progression to differing degrees in these different tumor models. Indeed, TMZ strongly reduced GL261 tumors as compared to the vehicle-treated group, which was significant from day 28 (Figure 2A). Similarly, TMZ also strongly reduced U87MG tumors as compared to vehicle-treated group, which was significant from day 26 (Figure 2B) or day 24 (Appendix A). Both GL261 and U87MG are highly responsive to TMZ, consistent with their *MGMT* hypermethylated profile, which favors TMZ sensitivity [40,41,42,43,44,47,48,49,50]. Conversely, TMZ slightly reduced U118MG tumor growth significantly at day 23 (Figure 2C) or at day 18 (Appendix A), but TMZ did not significantly affect U138MG tumor growth (Figure 2D and Appendix A). Indeed, both U118MG and U138MG have been described as having a low [43,45,46] and unmethylated [43,44] *MGMT profile*, respectively, suggesting poor response to TMZ, consistent with our observations.

### 3.3. Temozolomide Reduced GBM Tumor Growth in Syngenic and Xenograft Orthopic Models Associated with Acquired Resistance

In order to evaluate the TMZ response in a more relevant context, we also evaluated the response to TMZ using drug-sensitive tumors in the GL261 orthotopic allograft and U87MG xenograft models. Both cell lines have been described to express S1PRs [64,65,66,67,68]. This time, both cell lines were injected intracerebrally into the mice striatum. We used reporter cells expressing luciferase coupled with a bioluminescence imaging system in order to monitor the tumor growth in situ. We demonstrated in the GL261 orthotopic allograft that TMZ is particularly efficient and reduced tumor burden until day 21 (Figure 3A), similarly to what we observed in the subcutaneous model. Then, the tumor started to regrow starting from day 23, which led to a similar bioluminescence signal at day 43 (Figure 3A). We also monitored mouse body weight and survival in tumor-bearing mice upon treatment in sham and naïve mice. We observed an impact on morbidity with a reduction in body weight in parallel with an increase in bioluminescence signal, which was associated with mouse mortality events (Figure 3B,C). Similarly, we demonstrated again that U87MG-RFP-Luc tumors are also particularly sensitive to TMZ treatment with a significant reduction in tumor burden from day 20 (Figure 4A), and regrowth from day 30, which lead to similar bioluminescence at day 53 (Figure 4A). As in the GL261 model, U87MG-tumor-bearing mice also displayed morbidity and mortality which were significantly reduced upon TMZ treatment (Figure 4B,C). These findings suggest a resistance mechanism occurring which is consistent with TMZ response in patients [13]. These models somehow mimic the drug profile in positive responder patients, displaying a transient effect of TMZ on GBM tumors.

### 3.4. Analysis of Public Clinical Data: Expression of Downstream Target of Fingolimod Demonstrating That S1PR4 Is Correlated with Worse Prognostic Value in TCGA GBM Patient Cohort

Even if TMZ could have a transient anti-tumoral effect in the GBM model or in patients, ultimately it fails to avoid tumor recurrence and patient death [13], as observed in vivo (Figure 3). It therefore remains important to evaluate new targeting strategies in GBM. Recently, lipid metabolism in the brain, such as sphingolipid metabolism dysregulation, has attracted high interest both in normal and pathological situations, including GBM. Indeed, an aberrant sphingolipid metabolism is active in GBM and participates in tumor malignancy and progression [69]. Moreover, sphingosine kinases and S1P are described to be upregulated in GBM [65]. We therefore analyzed publicly available datasets of patients with GBM. We analyzed the expression level of S1P receptors, i.e., S1PR1, S1PR2, S1PR3, S1PR4, and S1PR5 [27], in the TCGA-GTEx GBM patient cohort. We showed that S1PR1 and S1PR5 expression levels are not affected in GBM tumors when compared to non-tumoral brains from the TCGA cohort (Figure 5A,E). Conversely, S1PR2, S1PR3, and S1PR4 are upregulated, which is significant for S1PR2 and S1PR3 (Figure 5B–D). We also evaluated patient survival in this cohort. Patients were stratified into groups expressing S1PRs at a low or high level using the median as the cut-off value. We observed first that a high expression of S1PR1, S1PR2, S1PR3, and S1PR5 did not significantly affect patient survival (Figure 5F–H,J). Conversely, S1PR4 is significantly correlated with a lower OS for GBM patients (Figure 5I). Moreover, the analysis of the LGG cohort demonstrated a similar profile to S1PR1, S1PR2, S1PR3, and S1PR4, which are upregulated in tumor tissue, significantly so for S1PR2 and S1PR3 (Appendix A–E). Furthermore, we also observed that a high expression of S1PR3 is significantly correlated with limited LGG patient survival (Appendix A–J). Interestingly, Fingolimod is described to antagonize S1PR4, as well as S1PR1, S1PR3, and S1PR5 [70]. Furthermore, previous research also described an overexpression of S1PR1 and S1PR3 and poor prognosis for S1PR1 in GBM patients in an alternative cohort [65,71]. Altogether, these data highlight that targeting the S1P axis might be valuable as a therapeutic strategy for glioma/GBM patients, suggesting the potential of S1PR inhibitors such as Fingolimod [27].

### 3.5. Fingolimod Induced Cytotoxicty in 2D Culture but Weakly Affected 3D Spheroid Culture

In order to analyze the effect of Fingolimod on GBM, we evaluated the Fingolimod dose–response relationship in GBM cell lines in 2D and 3D settings, as we did for TMZ (Figure 1 and Appendix A). We demonstrated that Fingolimod induced a dose–response increase in 2D cytotoxicity which is significant for the dose of 1 µM for both U87MG (Appendix A) and U118MG cells (Appendix A). These data are consistent with previous work, which also demonstrated the ability of Fingolimod to reduce GBM cell viability in 2D culture at similar doses [33,34,35,72]. Conversely, using 3D spheroid models, we demonstrated that Fingolimod did not affect U87MG spheroid growth (Figure 6A) nor cytotoxicity, even if a minimal increase might be observed at 10 µM (Figure 6A,B). Similarly, Fingolimod did not affect the growth or cytotoxicity of both U118MG and U138MG spheroid models (Appendix A–F). Fingolimod also did not significantly affect GL261 spheroid growth (Figure 6D) but induced a limited but significant increase in 3D cytotoxicity at the highest dose of 10 µM starting from 57 h (Figure 6E,F). The Fingolimod response varies strikingly between 2D and 3D models, with an almost absence of Fingolimod anti-tumor effect in 3D spheroid assays. The use of 3D models has been shown to better mimic the drug response and resistance to anti-cancer drugs [73,74,75]. Indeed, 3D models have generally shown less drug sensitivity and have already been used for GBM models [76]. This may explain the difference in Fingolimod response between the 2D and 3D assays.

### 3.6. Fingolimod Did Not Affect GBM Progression In Vivo in Immunocompetent and Immunodeficient Orthotopic Mouse Models

We finally evaluated the response of Fingolimod as a monotherapy or in combination with TMZ in vivo in a relevant orthotopic context, using both drug-sensitive tumors, GL261 allograft and U87MG xenograft models. In this assay, an additional sub-optimal dose of TMZ was used in order to visualize the potential synergistic effect of Fingolimod in co-treatment. We confirmed first that a high dose of TMZ at 10 mg/kg is particularly efficient at significantly reducing tumor growth from day 17 (Figure 7A). Conversely, the lowest dose of TMZ at 1 mg/kg did not significantly affect tumor growth even if a transient reduction could be observed at days 10 and 15, suggesting rapid drug resistance (Figure 7A). Fingolimod did not affect the tumor kinetics, alone or in combination, with TMZ at a low dose (Figure 7A). We also monitored mouse morbidity, focusing on body weight and neuroscore evaluation, using tumor-bearing mice upon treatment and sham mice. Naïve and sham mice demonstrated similar responses in our previous assay with GL261 tumors (Figure 3B,C). We observed that all groups, except for the high-dose TMZ-treated and sham mice, exhibited body weight loss concomitantly with tumor progression (Figure 7B). The same mice also displayed a decrease in neuroscore as compared to the high dose of TMZ-treated and sham mice, which was only significant versus sham mice (Figure 7C). These data suggest that the high dose of TMZ reduced the morbidity of mice. Moreover, the high dose of TMZ also prevented the mortality of mice (Figure 7D). 

Fingolimod response was also evaluated in the U87MG xenograft model. We also confirmed that a high dose of TMZ at 10 mg/kg reduced tumor progression (Figure 8A). Interestingly, a lower dose of TMZ at 1 mg/kg also reduced tumor growth in a weaker manner, demonstrating a dose-dependent response (Figure 8A). Moreover, both doses of TMZ reduced morbidity with an absence of body weight loss as compared to sham mice and reduced clinical symptoms as evaluated by the neuroscore, even if not significant (Figure 8B,C). TMZ also prevented mouse mortality (Figure 8D). Conversely, Fingolimod as compared to the control did not affect tumor growth, did not prevent body weight loss, did not improve neuroscore, and did not prevent mouse mortality. These data show that Fingolimod did not influence tumor progression, morbidity, or mortality (Figure 8A–D). Indeed, even when combined with the low dose of TMZ, Fingolimod and TMZ reduced tumor growth but did not enhance the anti-tumoral effect observed with TMZ alone, suggesting an absence of a synergistic effect between the two drugs (Figure 8A). Starting from day 30, several signs of morbidity were observed in the groups treated with vehicles (control), Fingolimod, or a combination that led to the sacrifice of some mice; we stopped the monitoring at this timepoint (Figure 8D).

## 4. Discussion

In this work, we evaluated the TMZ response for up to four GBM cellular models. This is the first time, to the best of our knowledge, that a comparison has been made in one study of the effects of TMZ in a TMZ-responsive tumor model, i.e., U87MG and GL261 cells, and of non-responsive models related to their MGMT promotor methylation status, i.e., U118MG and U138MG cells. TMZ is the only FDA-/EMA-approved drug for GBM, demonstrating a limited but significant effect on OS [4]. This is also the first study comparing the potential anti-tumoral response of Fingolimod using these four GBM cell lines. We hope these data will help to better define the specific interest of each model for the evaluation of new drugs. Indeed, we demonstrated that *MGMT* promotor methylation status is associated with treatment efficacy. Both hypermethylated models, U87MG and GL261, were sensitive to TMZ treatment even if acquired resistance appeared (Figure 2, Figure 3 and Figure 4), as observed in the clinic [13]. Conversely, both of the low and unmethylated models, U118MG and U138MG, demonstrated a limited response to TMZ (Figure 2).

Tumor cell grafts have many advantages, including effective and rapid tumor growth, for the evaluation of tumor biology and new therapy. Several authenticated cell lines can be used and injected into mice brains by stereotaxic intracranial or subcutaneous injection, while only intracranially injected tumors can mimic the GBM tumor microenvironment [77]. The allograft model, using cells such as the mouse GL261 cell line, permits the analysis of GBM progression and infiltration in a relevant immune microenvironment. GL261 is a well-characterized model displaying multiple features of GBM, including high inflammation, angiogenesis, high levels of major histocompatibility complex class, high CXCR4/CXCL12 and Ras expression, and p53 tumor suppressor gene point mutations [16]. Nevertheless, other cell lines have been generated over the years, such as GL26 or CT-2A, displaying alternative, interesting features [77]. The evaluation of immunomodulatory molecules can be feasible in such models that could also be used in C57BL/6J background transgenic mice to further study the specific role of immune cells. Indeed, the GL261 orthotopic model has already demonstrated that anti-PD-1 combined with TMZ promoted mouse survival [78]. Nevertheless, the syngeneic models cannot replicate the human immune system. Thus, xenograft models using human cells often remain essential to evaluate biologics. Xenograft models use immunodeficient mice such as nude, severe combined immunodeficient (SCID), or non-obese diabetic severe combined immunodeficiency (NOD/SCID) mice. All strains lack some features of the normal immune microenvironment. Nevertheless, recent progress in modeling the human immune system in mice may unravel such issues, making cell engraftment still a good option to evaluate new molecules by using TMZ-sensitive cell lines, such as U87MG, or TMZ-resistant cell lines, such as U118MG and U138MG [77].

The additional aim of this study was to evaluate the potential anti-tumor effect of Fingolimod in GBM models. Fingolimod displays immunomodulatory functions and is used as a treatment for multiple sclerosis. Indeed, Fingolimod induces peripheral lymphocyte sequestration, causing them to be retained in secondary lymphoid organs and limiting brain infiltration [79]. Fingolimod administered orally displays a good bioavailability and it penetrates into the brain through the blood–brain barrier [32]. Fingolimod is reversibly phosphorylated by sphingosine kinases into its water-soluble variant, Fingolimod phosphate. Multiple sclerosis patients are treated with 0.5 mg per day, with a good safety profile [32]. At this dose, Fingolimod reaches a blood concentration varying between 10 and 100 ng/mL [80,81]. In our study, Fingolimod was used based on this posology, but a higher concentration could be achieved by increasing the daily dose [82]. Fingolimod and Fingolimod phosphate have a low renal clearance associated with a drug half-life of around 6–9 days and thus display stable concentrations in the bloodstream [81]. Nevertheless, only one not-yet-peer-reviewed clinical study has been carried out in the GBM context, but no data on efficacy were evaluated [83].

In recent years, has Fingolimod demonstrated an anti-cancer effect in multiple tumor models in vitro and in vivo [25,27], but the mechanisms behind its anti-cancer effect are not yet fully characterized. We and others have demonstrated that Fingolimod can induce in vivo anti-tumor effects in triple-negative breast cancer, hepatocellular carcinoma, prostate cancer, and lung cancer models [28,29,84,85]. Moreover, similar responses were observed in both immunocompetent and immunodeficient mice. The absence of functional T cells in immunocompetent mice suggests that the anti-tumor effect of Fingolimod is a T-cell-independent mechanism [28,29]. In order to investigate the GBM response upon treatment with Fingolimod in combination with TMZ, we used in vitro assays as well as in vivo human and mouse orthotopic models. We demonstrated that although Fingolimod reduced GBM in vitro 2D cell growth, it did not globally affect 3D cell growth and in vivo progression. No synergistic effect has been observed with the combination of TMZ and Fingolimod in our GBM models. In the literature, Fingolimod has been already evaluated in rat 9L gliosarcoma and C6 glial tumor models. C6 cells are derived from glial tissue and do not express IDH1 [86], such that they are not classified as GBM. In addition, 9L gliosarcoma cells are not classified as GBM [77]. In both models, Fingolimod reduces tumor burden [36]. Moreover, Fingolimod also decreases tumor burden and improves survival in orthotopic xenograft models using brain tumor stem cells [38,87]. Finally, one research group showed the anti-tumoral effect of Fingolimod on U87MG and U251MG GBM cells, but the cells were injected subcutaneously and not into the brain [35], which is a different, poorly relevant microenvironment. Here, we showed the first data analyzing the effect of Fingolimod in human orthotopic GBM models and did not confirm the anti-tumor response described in the previous literature, even if Fingolimod crosses the blood–brain barrier and interacts with tumors [32,88].

The tumor microenvironment influences cancer progression through different mechanisms, including neo-angiogenesis. This critical step in tumor development induces the formation of neo-vessels within the tumor, favoring its progression [89]. Interestingly, S1P signaling affects cancer progression tumor angiogenesis, and is highly expressed in GBM, suggesting a potential role in brain tumor angiogenesis [27,90]. TMZ has not been described to affect tumor vasculature, whereas Fingolimod affects tumor angiogenesis through the inhibition of the S1P receptor axis [27,91]. Indeed, Fingolimod reduced in vitro angiogenesis by reducing the migration/invasion of endothelial cells while not affecting tube formation [92,93,94,95]. The effect of Fingolimod is quite versatile depending on the model used, since in the brain ischemia model, Fingolimod also promotes neo-angiogenesis [94]. Fingolimod can reduce blood vessel density in breast and lung cancer models, reducing tumor growth but also inducing tumor normalization [27,92,93]. Tumor normalization can thus promote vessel perfusion and enhance drug delivery and efficacy into the tumor, which is generally associated with a better effect of treatment given in combination [92]. In our work, we did not identify either effect of Fingolimod, since Fingolimod by itself did not affect tumor progression, whereas in combination it did not promote the effect of TMZ. To our knowledge, no further data have been published on the effect of Fingolimod in GBM tumor angiogenesis in the brain context. Thus, the effect of Fingolimod in the brain TME remains unclear and, based on our data, more research is required.

Interestingly, we demonstrated that Fingolimod at similar doses affects 3D culture and 2D culture differently (Appendix A, Figure 6, and Appendix A). Globally, the cytotoxic effect observed in 2D culture is not noticed in 3D spheroid models, which better recapitulate the tumor in situ situation [76,96]. Consistent with 3D in vitro models, we did not observe any in vivo anti-tumor effect of Fingolimod, contrary to data described in other tumors, such as hepatoma, non-small cell lung carcinoma, and triple-negative breast cancer [28,29,97]. The difference in anti-tumor response between our 2D and 3D systems showed that 3D models are less sensitive to Fingolimod treatment. Despite the fact that 2D models still represent the standard for evaluating cellular behavior in vitro, this approach could reveal important limitations to extrapolate in situ responses [17]. Conversely, 3D models display cell–matrix interactions, cell morphology, molecular signatures, and environmental gradients that better mimic in situ cellular functions [96,98,99]. Fingolimod affects different processes such as tissue remodeling, tumor angiogenesis, the immunomodulatory response, or hypoxia in other tumor indications [25,100]. Moreover, although Fingolimod is described to sensitize or enhance the drug response in vitro on GBM or lung cancer cells [39,101], no synergistic effect of Fingolimod and TMZ was observed in vivo. Interestingly, a recent clinical trial evaluating the safety profile of Fingolimod in GBM-treated patients (NCT02490930) suggested a good tolerability of Fingolimod even if no additional anti-tumoral effect was described (preprint available at this address: https://doi.org/10.21203/rs.3.rs-251903/v1, accessed 5 July 2023). One could therefore speculate that some compensatory mechanisms may affect the Fingolimod response and limit its anti-tumor effect in the tumor microenvironment. The lack of a defined molecular mechanism or factors involved in the differential drug response observed upon Fingolimod treatment is a limitation of our study. Additional work is required in order to elucidate how Fingolimod modulates the molecular pathways responsible for the cytotoxic effect in 2D in the absence of consistent effects in 3D in vitro or in vivo situations. A focus on the downstream targets of Fingolimod in GBM cells identified in other tumor indications, for instance, could be meaningful, but was not investigated here.

Nevertheless, further evaluation of the S1P inhibitor might still be relevant for glioma patients and is being investigated as a new target for glioma [69]. Indeed, using a publicly available TCGA dataset, we observed that S1PR2 and S1PR3 are significantly overexpressed in GBM, but no correlation with patient OS for GBM was observed. Meanwhile, S1PR3 is correlated with poor patient OS for LGG, but not for GBM patients (Figure 5 and Appendix A), and S1PR4 is non-significantly overexpressed in LGG and GBM but is correlated with poor patient OS for GBM (Figure 5 and Appendix A). Nevertheless, only S1PR4 is correlated with GBM patient survival, despite some trends for other receptors. Interestingly, other authors also confirmed that S1PR1 and S1PR3 are upregulated in GBM patients and that S1PR1 is correlated with patient survival in an alternative database [65]. Fingolimod targets S1PRs, including S1PR1, S1PR3, S1PR4, and S1PR5, but does not affect S1PR2 [24,25]. Therefore, alternative drugs targeting specific S1PRs might be relevant for glioma and glioblastoma. Additionally, recent data have suggested that targeting S1PRs in a more specific manner, such as S1PR2 and S1PR4 blockade, displayed an anti-tumoral effect or limited chemotherapy resistance in other tumor indications [102,103,104].

## 5. Conclusions

In conclusion, we demonstrated the predictive value of the evaluated GBM models in this study in response to TMZ. We also demonstrated that Fingolimod did not display 3D in vitro or in vivo anti-cancer effects, whereas it affects GBM cells cultured in 2D in vitro. This study raised the question of the potential of Fingolimod and its therapeutic value at the preclinical level. We trust that these encouraging data will stimulate the glioma research community to further evaluate the potential of targeting the S1P/S1PRs axis in brain tumors.

## Figures and Tables

**Figure 1 cancers-15-04478-f001:**
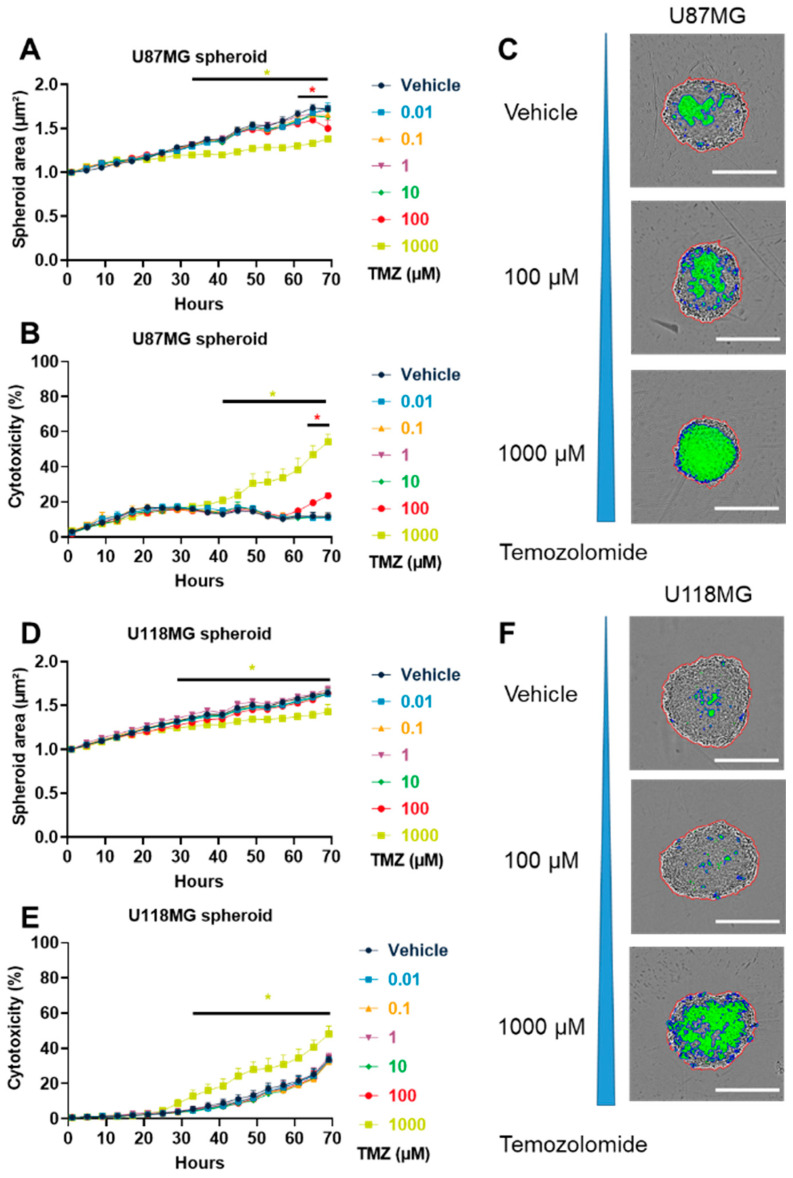
Effects of Temozolomide (TMZ) on glioblastoma cells in 3D tumor spheroid assay. (**A**–**F**) Dose–response effect of TMZ treatment on U87MG (**A**–**C**) and U118MG (**D**–**F**) spheroid size (**A**,**D**) and cytotoxicity (**B**,**E**). The spheroid area (red line) and fluorescent-positive surface (blue line) were quantified per spheroid (**C**,**F**). Representative pictures of cells after 69 h of post-treatment with 0, 100, and 1000 µM of TMZ are shown. Scale = 500 µm. The assay was performed with 3 to 4 replicates from 2 to 3 independent experiments. Data represent the mean and SEM. Statistical differences were determined using a mixed-effects model (REML, groups and time as factor) and Bonferroni’s multiple comparisons test (vs. control, * *p*  ≤  0.05).

**Figure 2 cancers-15-04478-f002:**
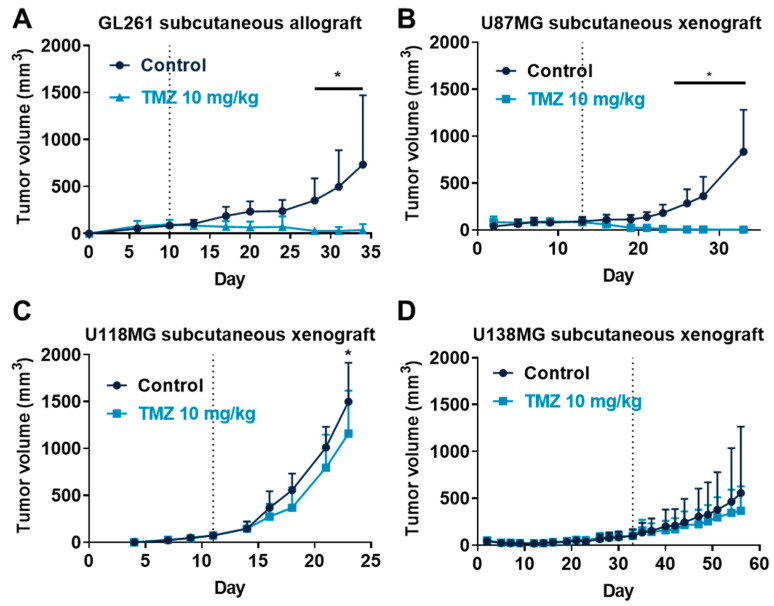
Effects of Temozolomide (TMZ) treatment in glioblastoma mouse subcutaneous graft models. (**A**–**D**) Impact of TMZ at 10 mg/kg administered 5 times a week (p.o.) on tumor volume in GL261 allograft (**A**), U87MG xenograft (**B**), U118MG xenograft (**C**), and U138MG xenograft (**D**) models. Discontinuous line highlights treatment beginning. Statistical differences between the groups were determined using a mixed-effects model (REML, groups and time as factor) followed by Bonferroni’s multiple comparisons test (* *p*  ≤  0.05). Data represent mean and SD. n = 10 (**A**), n = 6 (**B**), n = 7 (**C**), and n = 8 (**D**) mice per group at the start of treatments.

**Figure 3 cancers-15-04478-f003:**
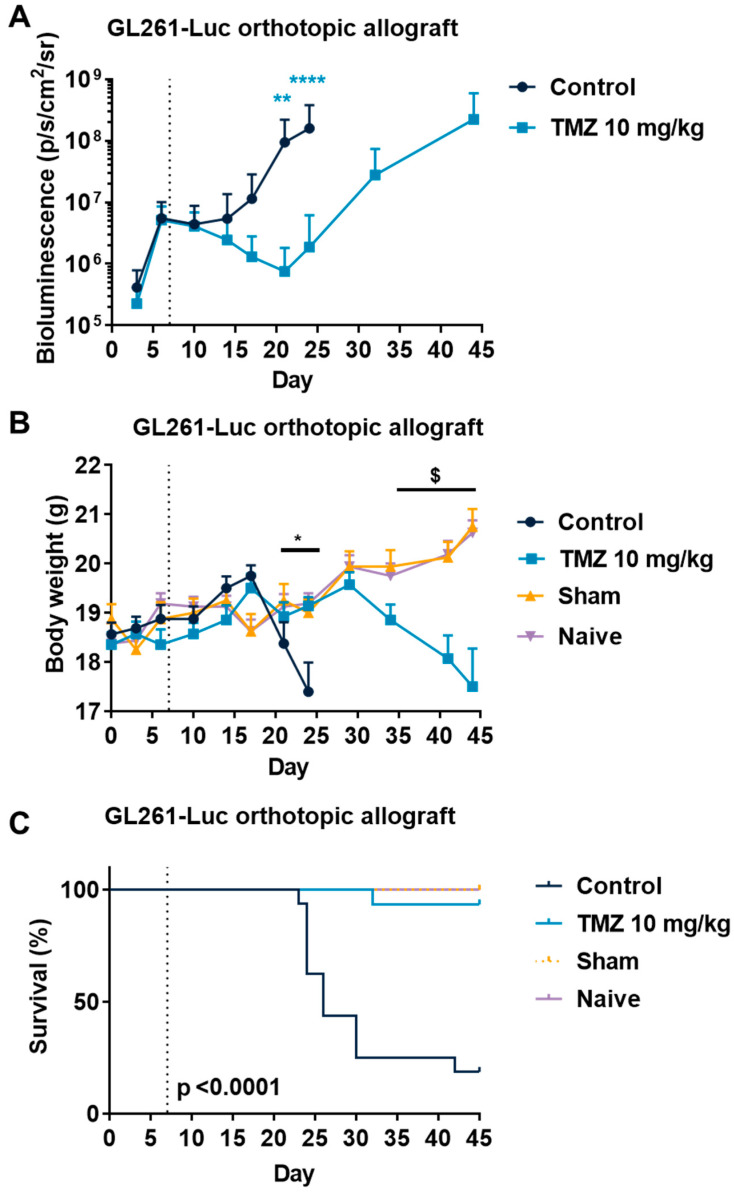
Effects of Temozolomide (TMZ) treatment in the GL261 glioblastoma mouse orthotopic allograft model. (**A**–**C**) Impact of TMZ at 10 mg/kg administered 5 times a week (p.o.) on tumor volume (**A**), body weight (**B**), and survival (**C**). Discontinuous line highlights treatment beginning. Statistical differences between the groups were determined using a mixed-effects model (REML, groups and time as factor) followed by Bonferroni’s multiple comparisons test (* *p*  ≤  0.05, ** *p*  ≤  0.01, **** *p*  ≤  0.001, control vs. all other condition; $ *p*  ≤  0.05, TMZ 10 mg/kg vs. sham and naïve). Data represent mean and SD. n = 8 mice per group at the start of treatments. For body weight, data reporting was stopped when the first mouse per group had to be sacrificed.

**Figure 4 cancers-15-04478-f004:**
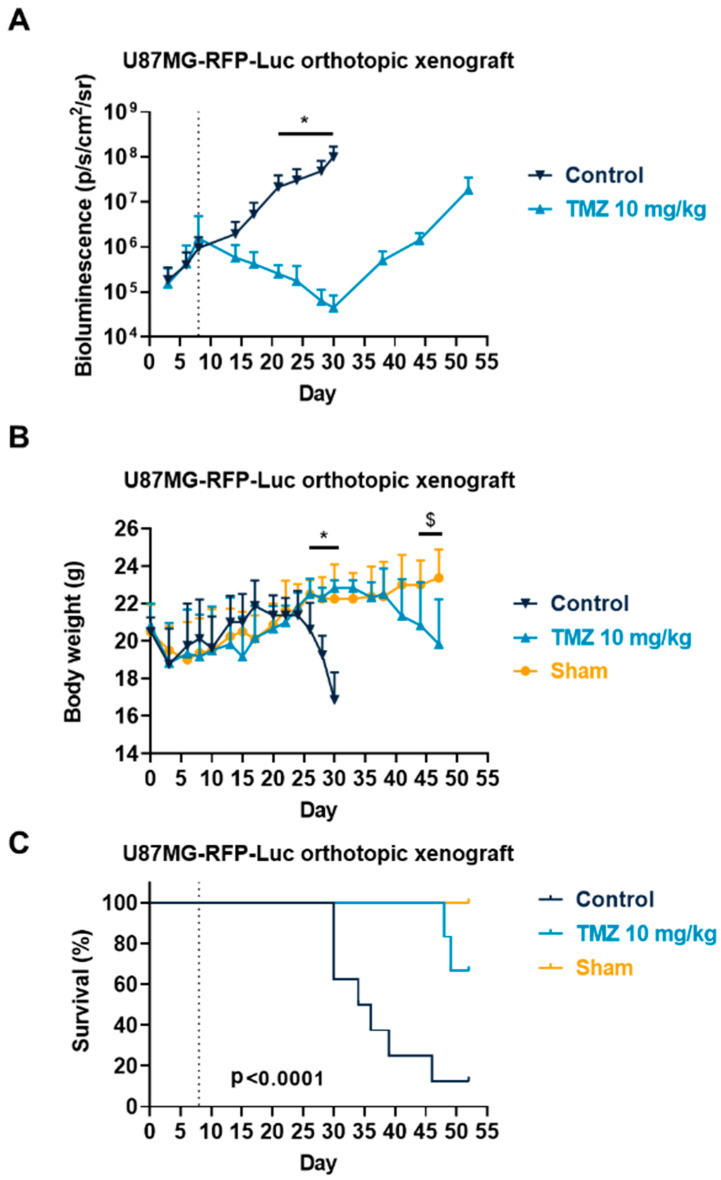
Effects of Temozolomide (TMZ) treatment in the U87MG-RFP-Luc glioblastoma mouse orthotopic xenograft model. (**A**–**C**) Impact of TMZ at 10 mg/kg administered 5 times a week (p.o.) on tumor volume measured by bioluminescence imaging (**A**), body weight (**B**), and survival (**C**). Discontinuous line highlights treatment beginning. Statistical differences between the groups were determined using a mixed-effects model (REML, groups and time as factor) followed by Bonferroni’s multiple comparisons test (* *p*  ≤  0.05, control vs. all other condition; $ *p*  ≤  0.05, TMZ 10 mg/kg vs. Sham). Data represent mean and SD. n = 8 mice per group at the start of treatments. For body weight, data reporting was stopped when the first mouse per group had to be sacrificed.

**Figure 5 cancers-15-04478-f005:**
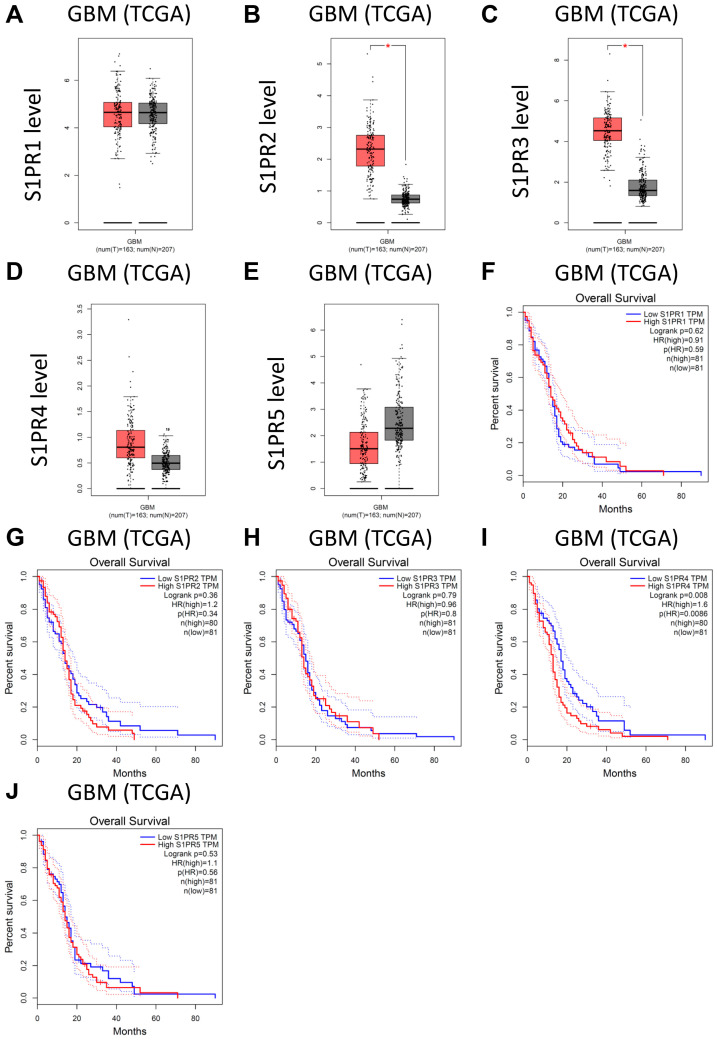
Analysis of publicly available clinical data of S1PR1, S1PR2, S1PR3, S1PR4, and S1PR5 expression in GBM patients from the TCGA cohort. (**A**–**E**) S1PR2 and S1PR3 are significantly overexpressed in GBM tumors as compared to non-tumoral brains. S1PR1, S1PR4, and SP1R5 are not significantly affected despite an observable increase for S1PR4. The number of patients in each group is indicated below the graphs (T = tumor, in red; N = normal non-tumoral brain, in grey). The difference in expression level between the groups of individuals was evaluated by ANOVA. * *p*  ≤  0.05 (**F**–**J**) High expression of S1PR4 is correlated with more limited patient survival. Expression of S1PR1, S1PR2, S1PR3, and S1PR5 did not correlate with patient survival. Patients were stratified through high and low expression of S1PRs using the median as the cut-off value. The number of patients in each group is indicated. The statistical difference in survival was evaluated using the log-rank test.

**Figure 6 cancers-15-04478-f006:**
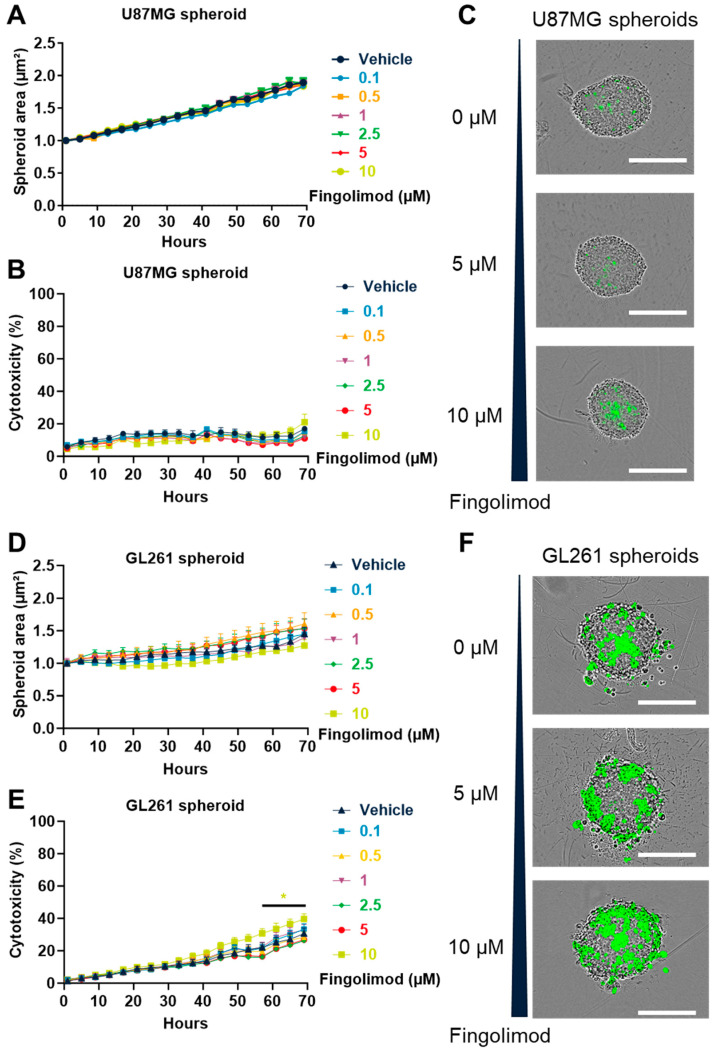
Effects of Fingolimod on GBM cells in 3D tumor spheroid assay. (**A**–**F**) Dose–response effect of Fingolimod treatment on U87MG (**A**–**C**) and GL261 (**D**–**F**) spheroid size (**A**,**D**) and cytotoxicity (**B**,**E**). The spheroid area and fluorescent-positive surface were quantified per spheroid. Representative pictures of cells after 69 h of post-treatment with 0, 5, and 10 µM of Fingolimod are shown. Scale = 500 µm. The assay was performed with 3 to 4 replicates from 2 to 3 independent experiments. Data represent mean and SEM. Statistical differences were determined using a mixed-effects model (REML, groups and time as factor) and Bonferroni’s multiple comparisons test (vs. control, * *p*  ≤  0.05).

**Figure 7 cancers-15-04478-f007:**
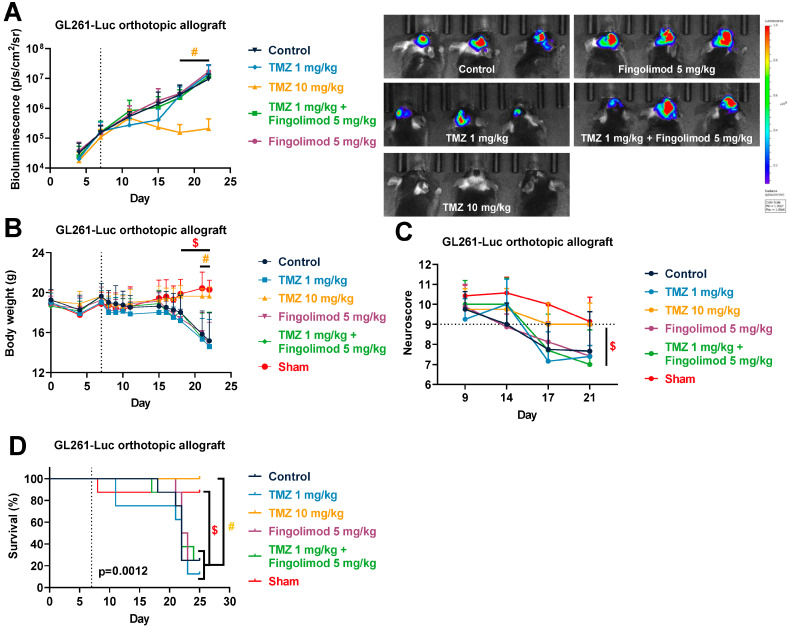
Effects of Temozolomide (TMZ) and Fingolimod treatments in the GL261 glioma mouse orthotopic allograft model. (**A**–**D**) Impact of TMZ at 1 and 10 mg/kg, Fingolimod at 5 mg/kg, or a combination administered 5 times a week (p.o.) on tumor volume measured by bioluminescence imaging, including representative images (**A**), body weight (**B**), neuroscore (**C**), and survival (**D**). Discontinuous line highlights treatment beginning. Statistical differences between the groups were determined using a mixed-effects model (REML, groups and time as factor) followed by Tukey’s multiple comparisons test # *p*  ≤  0.05, TMZ 10 mg/kg vs. Control or TMZ 1 mg/kg or TMZ 1 mg/kg + Fingolimod 5 mg/kg or Fingolimod 5 mg/kg; $ *p*  ≤  0.05, Sham vs. Control or TMZ 1 mg/kg or TMZ 1 mg/kg + Fingolimod 5 mg/kg or Fingolimod 5 mg/kg). Data represent mean and SD. n = 8 mice per group at the start of treatments.

**Figure 8 cancers-15-04478-f008:**
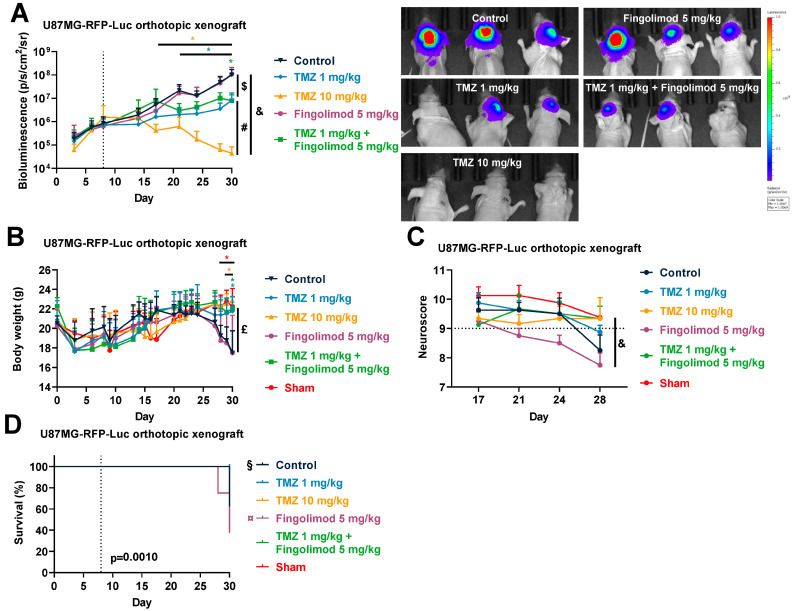
Effects of Temozolomide (TMZ) and Fingolimod treatments in the U87MG human orthotopic xenograft model. (**A**–**D**) Impact of TMZ at 1 and 10 mg/kg, Fingolimod at 5 mg/kg, or a combination administered 5 times a week (p.o.) on tumor volume measured by bioluminescence imaging, including representative images (**A**), body weight (**B**), neuroscore (**C**), and survival (**D**). Discontinuous line highlights treatment beginning. Statistical differences between the groups were determined using a mixed-effects model (REML, groups and time as factor) followed by Tukey’s multiple comparisons test (* *p*  ≤  0.05, vs. Control; $ *p*  ≤  0.05, Fingolimod 5 mg/kg vs. TMZ 1 mg/kg or TMZ 1 mg/kg + Fingolimod 5 mg/kg; & *p*  ≤  0.05, Fingolimod 5 mg/kg vs. Sham; £ *p*  ≤  0.05, Fingolimod 5 mg/kg vs. TMZ 1 mg/kg or TMZ 1 mg/kg + Fingolimod 5 mg/kg or TMZ 10 mg/kg or Sham; # *p*  ≤  0.05, TMZ 10 mg/kg vs. TMZ 1 mg/kg or TMZ 1 mg/kg + Fingolimod 5 mg/kg; ¤ *p*  ≤  0.05, Control vs. TMZ 10 mg/kg or TMZ 1 mg/kg or TMZ 1 mg/kg or TMZ 1 mg/kg + Fingolimod 5 mg/kg or Sham; § *p*  ≤  0.05, Fingolimod 5 mg/kg vs. TMZ 10 mg/kg or TMZ 1 mg/kg or TMZ 1 mg/kg or TMZ 1 mg/kg + Fingolimod 5 mg/kg or Sham). Data represent mean and SD. n = 6–8 mice per group at the start of treatments.

## Data Availability

The data presented in this study are available on request from the corresponding author. Any request should be addressed to Dr. Tristan Rupp, rupptristan@hotmail.fr.

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
