# Peer review of "Evaluation of Temozolomide and Fingolimod Treatments in Glioblastoma Preclinical Models"

_cancers, 2023, doi:10.3390/cancers15184478_

Round 1
Reviewer 1 Report
The manuscript: “Evaluation of Temozolomide and Fingolimod treatments in gli- 2 oblastoma preclinical models” is focused on the evaluation of the effect of Fingolimod on different glioma models. The topic is relevant because novel GBM cures are needed as wll as a better understand of mechanism of TMZ resistance. Despite the interest several part of the manuscript should be carefully revised according the points listed above because as presented the manuscript is not conclusive.
Introduction: some pharmakokinetics information on Fingolimod should be addressed to better address the rational of its use in GBM
Design: Subcunaneous and orthotopic models have been evaluated. Subcutaneous models does not represent GBM however a potential rational should be the use of these model to overcome the unknown brain GBM penetrabilty of Fingolimod. However subcutaneius models were tested only for TMZ alone and not for drug combination. For this reason the rational of subcunaneous model is not clear (selection of TMZ responce not yet evaluated?). Another point is the selection of in vivo TMZ dose and the design for Survival Curves and analysis particularly that describe in Fig. 8. Why, the effect of Fingolimod associated with the highest effective dose of TMZ and the evaluation of a potential increase in survival was not tested?.
Line 61. Please modify resistance in: “resistance or acquired resistance” since some lesions show recurrence after an initial responce to RT plus temozolomide treatment.
Line 75: Limits of in vivo models based on immunodeficent mice that the lack a complete immune responce should be also addressed.
Methods
General: please explain the rationale of using subcutaneous models to test TMZ in vivo. In most of the models evaluated TMZ efficacy has been already tested in the orthotopic one.
Line 189: Please indicate the rationale of using a 10mg/kg TMZ dose particularly in comparison with the clinical regimen used in patients.
Line 209: Please correct subcutaneous with orthotopic
Line 391: Why the targets identifided from data base were not evaluated tested in the orthotopic mice models?
Line 428: “High expression of S1PR4 is correlated with poor overall survival (OS) of GBM patients”. The effect is really low. Please comment it in the discussion section
Figure 8, panel D. Mortality look the same between the evaluated groups except for Fingolimod alone. Please comment. Were animals sacrificed on the basis of clinical sign or after 30 days independently? Kaplan Mayer curve is not clear.
Discussion
Intratumoral uptake of diffusion withing neurospheres of Fingolimod might affect its efficacy. Please discuss carefully the kinetics properties of Fingolimod and the presence in the model selected of sphingosine-1-phosphate receptors or activated pathway. Data on how and their expression or modulation by treatment given alone or in combination could give some more information on the involvement of sphingolipids in glioma.
Author Response
The manuscript: “Evaluation of Temozolomide and Fingolimod treatments in gli- 2 oblastoma preclinical models” is focused on the evaluation of the effect of Fingolimod on different glioma models. The topic is relevant because novel GBM cures are needed as wll as a better understand of mechanism of TMZ resistance. Despite the interest several part of the manuscript should be carefully revised according the points listed above because as presented the manuscript is not conclusive.
Authors’ response: We would like to thank first the reviewer for the very constructive comments on our manuscript and we hope that the response and new edit of the manuscript will convince the reviewer of the interest of our work.
Introduction: some pharmakokinetics information on Fingolimod should be addressed to better address the rational of its use in GBM
Authors’ response: As requested by the reviewer we included now further information with two paragraphs in both introduction and discussion sections describing the additional PK/PD information on Fingolimod demonstrating its interest in treatment of brain diseases (lines 98-101 and line 594-604).
Design: Subcunaneous and orthotopic models have been evaluated. Subcutaneous models does not represent GBM however a potential rational should be the use of these model to overcome the unknown brain GBM penetrabilty of Fingolimod. However subcutaneius models were tested only for TMZ alone and not for drug combination. For this reason the rational of subcunaneous model is not clear (selection of TMZ responce not yet evaluated?). Another point is the selection of in vivo TMZ dose and the design for Survival Curves and analysis particularly that describe in Fig. 8. Why, the effect of Fingolimod associated with the highest effective dose of TMZ and the evaluation of a potential increase in survival was not tested?.
Authors’ response: We thank the reviewer for this interesting comment.
Our goal was double, first to determine if Fingolimod by itself can reduce tumor growth as it did in preclinical breast and lung cancer models (Rupp et al., Transl Oncol. 2021 Jan;14(1):100926. / Rupp et al., Int J Mol Sci. 2022 Jul 25;23(15):8192), then to determine if Fingolimod could promote the effects of chemotherapy as it did in some models such as breast cancer (Mousseau et al., Breast Cancer Res Treat. 2012 Jul;134(1):31-40.). Thus, we decided to pick up a sub-optimal dose of TMZ in order to identify potential synergistic effect that cannot be achieved with the strong effect of TMZ by itself at high dose. We indeed did not evaluate Fingolimod and TMZ in a pure survival analysis. Actually, our goal was initially to evaluate potential mechanism of action of Fingolimod. Thus, we sacrificed mice at the same timepoint in order to potentially analyze biomarker modulation upon Fingolimod. Nevertheless, since no efficacy was observed we did not perform the measurement of the different biomarkers. Moreover, some signs of morbidity were observed, and some mice had to be sacrificed for ethical consideration in both models, pushing us to arrest the study.
Several teams have demonstrated that Fingolimod can cross the brain blood barrier (BBB) (Foster et al., J Pharmacol Exp Ther. 2007 Nov;323(2):469-75. / Hunter et al., CNS Drugs (2016) 30:135–147). Since we confirm that standard drug such as TMZ displayed a similar response in both subcutaneous and orthotopic context and since Fingolimod penetrates into the brain, we decided to go directly to the relevant orthotopic context. Indeed, as also mentioned by the reviewer, orthotopic model remain essential to better mimic the correct brain tumor microenvironment. Nevertheless, even if subcutaneous model does not mimic brain tumor microenvironment, subcutaneous models remain quite useful and straight forward method to identify first proof of concept of test substance’s efficacy.
Indeed, thanks to the evaluation of TMZ in subcutaneous models we identified the TMZ-sensitive models that was used for further evaluation of TMZ and Fingolimod in more relevant context with the intracerebral injection of tumor cells (orthotopic models).
Line 61. Please modify resistance in: “resistance or acquired resistance” since some lesions show recurrence after an initial responce to RT plus temozolomide treatment.
Authors’ response: Thanks for this comment, we edited accordingly.
Line 75: Limits of in vivo models based on immunodeficent mice that the lack a complete immune responce should be also addressed.
Authors’ response: Thanks for this comment, we added a comment on this limitation demonstrating that syngeneic models such as the GL261 model remains relevant for in vivo drug evaluation (lines 75-80).
Methods
General: please explain the rationale of using subcutaneous models to test TMZ in vivo. In most of the models evaluated TMZ efficacy has been already tested in the orthotopic one.
Line 189: Please indicate the rationale of using a 10mg/kg TMZ dose particularly in comparison with the clinical regimen used in patients.
Authors’ response: Our aim was to use optimal and suboptimal dose of TMZ devoid of toxic effects. While the dosing in human falls generally around 150 mg/m2 (Stupp et al., N Engl J Med. 2005 Mar 10;352(10):987-96.), the corresponding dose in mouse is 50 mg/kg (based on Nair and Jacob. J Basic Clin Pharm. 2016;7(2):27‐31.). Nevertheless, we already observed that TMZ can be toxic at doses close to 50 mg/kg which induces rapid morbidity while at 10 mg/kg we did not (data not shown). Moreover, other authors used 10 mg/kg or lower doses demonstrating a good anti-tumoral response similar to higher dose and they also observed toxicity at higher dose (Segura-Collar et al., Neurooncol Adv. 2022 Jan-Dec; 4(1): vdac155. / Saha et al., J Immunother Cancer. 2020 May;8(1):e000345). Thus, we decided to use the dose of 10 mg/kg has highest dose in this study.
This information was added in the manuscript (lines 210-216)
Line 209: Please correct subcutaneous with orthotopic
Authors’ response: Thanks for identifying this mistake, corrected in the new version of the manuscript (line 234)
Line 391: Why the targets identifided from data base were not evaluated tested in the orthotopic mice models?
Authors’ response: We identified in publicly available dataset that S1PR3, and S1PR4 are upregulated in GBM and that SP1PR4 is correlated with limited patient survival. Quite consistently, other authors also confirmed that S1PR1, and S1PR3 are upregulated and that S1PR1 is correlated with patient survival in alternative database (Bien-Möller et al., Oncotarget. 2016 Mar 15; 7(11): 13031–13046.). Our both cellular models used and evaluated in vivo are described to express these receptors (Annabi et al., Mol Carcinog. 2009 Oct;48(10):910-9. doi: 10.1002/mc.20541. / Bernhart et al., Biochem Pharmacol. 2015 Jul 15; 96(2): 119–130. / Bien-Möller et al., Oncotarget. 2016 Mar 15; 7(11): 13031–13046. / Y Xiao · 2018, https://epub.ub.uni-greifswald.de/frontdoor/deliver/index/docId/3956/file/Doctoral+Thesis_yxiao-1.pdf / Marx et al., Cancers (Basel). 2019 Apr; 11(4): 569. / Bien-Möller et al., Cancers 2023, 15(17), 4273). This information has been added to the new version of the manuscript (line 384). Interestingly, Fingolimod is described to target S1PR1 and S1PR4 (Geffken and Spiegel. AdvBiolRegul. 2018 January; 67: 59–65.). Based on these data, we decided to go for a pharmacological approach to evaluate the interest of targeting S1PRs using Fingolimod. The cell lines evaluated were sensitizing to Fingolimod in 2D in vitro culture, suggesting the interest of S1P targeting. Nevertheless, some compensatory mechanisms act in more complex and relevant situation while no effects were observed in 3D culture nor in vivo.
Line 428: “High expression of S1PR4 is correlated with poor overall survival (OS) of GBM patients”. The effect is really low. Please comment it in the discussion section
Authors’ response: We observed that a significant correlation between S1PR4 and survival using the TCGA and GTEx database. We understand that the effect might be considered as low. Nevertheless, they are limited number of biomarkers that are associated with high OS difference for GBM patients in the literature (MacIaczyk et al., Int J Mol Sci. 2022 Jul; 23(13): 7474.). The most interesting one with conservative clinical results remains certainly the methylation of the MGMT’s promoter (Struve et al., Oncogene. 2020 Apr;39(15):3041-3055. / Morandi et al., BMC Cancer. 2010 Feb 18;10:48.). Nevertheless, it remains quite interesting to identify that S1PRs may be relevant as potential biomarkers of patient survival in GBM and may promote further research on the signaling pathway.
We edited the sentence to better reflect the message in the new version of manuscript (line 458).
Figure 8, panel D. Mortality look the same between the evaluated groups except for Fingolimod alone. Please comment. Were animals sacrificed on the basis of clinical sign or after 30 days independently? Kaplan Mayer curve is not clear.
Authors’ response: Indeed, after the 30-days monitoring several mice in the control, Fingolimod, and combination groups displayed signs of morbidity and were sacrificed for ethical reasons. In addition, the study was stopped in order to collect brain samples for further histological analysis. Indeed, our original purpose was to potentially work on molecular aspects affected by Fingolimod. As Fingolimod did not lead to any anti-tumor effect, we did not further evaluate the brain of mice. Indeed, in the scope of this manuscript the analysis of tumor upon Fingolimod was not really relevant.
We precise the apparition of morbidity that requested sacrifice of animals after day 30 in the new version of the manuscript from our point of view (lines 524-526).
Discussion
Intratumoral uptake of diffusion withing neurospheres of Fingolimod might affect its efficacy. Please discuss carefully the kinetics properties of Fingolimod and the presence in the model selected of sphingosine-1-phosphate receptors or activated pathway. Data on how and their expression or modulation by treatment given alone or in combination could give some more information on the involvement of sphingolipids in glioma.
Authors’ response:
We would like to thank the reviewer for pointing out this missing information from the original manuscript. Indeed, we demonstrated that U118MG and U87MG are sensitive to Fingolimod in 2D culture but not in other models. We also added 2D culture experiment with GL261 cells upon Fingolimod demonstrated 2D anti-tumor effect (New figure S6 panel C), this information was relevant and was missing from our manuscript. Thus, our cell lines are sensitive to Fingolimod suggesting a potential effect through S1P receptors. Moreover, U87MG cells are described to express all the five S1P receptors at different levels and in particular in orthotopic tumor microenvironment (Annabi et al., Mol Carcinog. 2009 Oct;48(10):910-9. doi: 10.1002/mc.20541. / Bernhart et al., Biochem Pharmacol. 2015 Jul 15; 96(2): 119–130. / Bien-Möller et al., Oncotarget. 2016 Mar 15; 7(11): 13031–13046. / Bien-Möller et al., Cancers 2023, 15(17), 4273). GL261 cells are also described to express S1PR1 and the other S1P receptors at different levels at least in vitro (Bien-Möller et al., Cancers 2023, 15(17), 4273 / Y Xiao · 2018, https://epub.ub.uni-greifswald.de/frontdoor/deliver/index/docId/3956/file/Doctoral+Thesis_yxiao-1.pdf / Marx et al., Cancers (Basel). 2019 Apr; 11(4): 569.). Nevertheless, we cannot exclude that Fingolimod-dependent mechanism observed in 2D culture are S1P receptors-related. Indeed, receptor-independent activity of Fingolimod has been identified via for example the induction of autophagy, the inhibition of histone deacetylases, or binding of protein complex such as inhibitor 2 of protein phosphatase 2A (Wang et al., Curr Med Chem. 2020 Jun; 27(18): 2979–2993.).
The information regarding S1PR expression has now been added to the new version of the manuscript (line 384).
Please see the manuscript in attachment.

Reviewer 2 Report
In this paper, authors investigated the effectiveness of treatments for glioblastomas. The study explored the effects of Temozolomide and Fingolimod, in different models of glioblastoma including in vitro 2D and 3D cell survival analysis and in vivo implanted models. Authors found Temozolomide had varying effects on different tumor types. Fingolimod did not display 3D in vitro or in vivo anti-cancer effect whereas it affects GBM cells cultured in 2D in vitro. Overall it’s a good descriptive paper.
But Here are my comments on the manuscripts
(1) Kindly consider enlarging the dimensions of the images and fonts within figures 1, 2, 5, 6, 7, and 8. The current sizes make it challenging to discern the content effectively.
(2) Regarding Figure 7, could you please include the bioluminescence imaging?
(3) Within this paper, the authors propose that targeting the S1P axis could potentially serve as a therapeutic approach for glioma. In addition to the S1P axis, have other mechanisms associated to the S1P axis been explored as potential contributors?
(3) Tumor microenvironment has critical impact on the no-response of treatment, especially the including heterogeneity in vascular micro-environment. Could authors discuss more on glioma vascular microenvironment
Please also cite the following papers:
Wei, X., Meel, M.H., Breur, M. et al. Defining tumor-associated vascular heterogeneity in pediatric high-grade and diffuse midline gliomas. acta neuropathol commun 9, 142 (2021). https://doi.org/10.1186/s40478-021-01243-1
Minor edits required
Author Response
In this paper, authors investigated the effectiveness of treatments for glioblastomas. The study explored the effects of Temozolomide and Fingolimod, in different models of glioblastoma including in vitro 2D and 3D cell survival analysis and in vivo implanted models. Authors found Temozolomide had varying effects on different tumor types. Fingolimod did not display 3D in vitro or in vivo anti-cancer effect whereas it affects GBM cells cultured in 2D in vitro. Overall it’s a good descriptive paper.
Authors’ response: We would like to thank first the reviewer for the very constructive comments on our manuscript and we hope that the response and new edit of the manuscript will convince the reviewer of the interest of our work.
But Here are my comments on the manuscripts
(1) Kindly consider enlarging the dimensions of the images and fonts within figures 1, 2, 5, 6, 7, and 8. The current sizes make it challenging to discern the content effectively.
Authors’ response: We thank the authors for this comment, we have enlarged the figure in the new version of the manuscript.
(2) Regarding Figure 7, could you please include the bioluminescence imaging?
Authors’ response: We included for both new figures 7 and 8 representative pictures of the bioluminescence signal in mice.
(3) Within this paper, the authors propose that targeting the S1P axis could potentially serve as a therapeutic approach for glioma. In addition to the S1P axis, have other mechanisms associated to the S1P axis been explored as potential contributors?
Authors’ response: Indeed, several signaling pathways known to promote tumor growth seem to be directly associated with S1P axis. Sphingosine Kinase has a structural analogy with sphingosine which will then activate signaling pathways including the PI3K/AKT/mTOR involved in cancer progression (Zhu et al., Oncol Rep. 2015 Mar;33(3):1257-63. / (Brunkhorst et al., Front Cell Neurosci. 2014; 8: 283.). Interestingly, recent research articles suggest a link between NRF2 and S1P axis (Zhang and Wang. Pharmacol Rep. 2017 Dec;69(6):1186-1193. / Awuah et al., Discov Oncol. 2022 Dec; 13: 94.). NRF2 is described to repress cacner progression through the binding of NRF2 and Keap1 retaining NRF2 at the cytoplasmic level and induces cell death by increasing oxidative stress (Saidu et al., Mol Cancer Ther. 2017 Mar;16(3):529-539.). Alternative signaling pathways such as NfkB are also described to be associated with S1P axis and could also serve as target for further pharmacological evaluation (White et al., Oncotarget. 2016 Apr 26;7(17):23106-27).
(3) Tumor microenvironment has critical impact on the no-response of treatment, especially the including heterogeneity in vascular micro-environment. Could authors discuss more on glioma vascular microenvironment
Authors’ response: We thank the reviewer for this pertinent recommendation. Indeed, the tumor microenvironment influences cancer progression through different mechanism including neo-angiogenesis. This critical step in tumor development induces the formation of neo-vessels within the tumor favoring its progression (Sharma et al., Neurooncol Adv. 2023 Jan-Dec; 5(1): vdad009.). Interestingly, S1P signaling is affecting cancer progression tumor angiogenesis, and is highly expression in GBM suggesting a potential role in brain tumor angiogenesis TMZ is not described to affect tumor vasculature (Mathivet et al., EMBO Mol Med (2017)9:1629-1645.) whereas Fingolimod is described to affect tumor angiogenesis (White et al., Oncotarget. 2016 Apr 26; 7(17): 23106–23127.). Indeed, Fingolimod reduced in vitro angiogenesis by reducing migration/invasion of endothelial cells while not affecting tube formation (Mousseau et al., Breast Cancer Res Treat. 2012 Jul;134(1):31-40. / Schmid et al., J Cell Biochem. 2007 May 1;101(1):259-70 / Shang et al., Brain Res. 2020 Jan 1;1726:146509 / Zhou et al., Front Pharmacol. 2020; 11: 59.). Moreover, the effect of Fingolimod is quite versatile depending on the model used. In brain ischemia model, Fingolimod also promoted neo-angiogenesis (Shang et al., Brain Res. 2020 Jan 1;1726:146509.). Fingolimod can reduce blood vessel density in breast and lung cancer model reducing tumor growth but also induced tumor normalization (Mousseau et al., Breast Cancer Res Treat. 2012 Jul;134(1):31-40. / Schmid et al., J Cell Biochem. 2007 May 1;101(1):259-70). Tumor normalization can thus promote vessel perfusion and enhance drug delivery and efficacy into the tumor which is generally associated with better effect of treatment given in combination (Mousseau et al., Breast Cancer Res Treat. 2012 Jul;134(1):31-40.). In our work, we do not identify one or the other effect of Fingolimod since Fingolimod by itself did affect tumor progression or in combination did not promote the effect of TMZ. From our knowledge no further work have been done on the effect of Fingolimod in GBM tumor angiogenesis in brain context. Thus, the effect of Fingolimod in the brain TME remains unclear and based on our data should request more research.
We added this information in the revised version of the manuscript (line 629-648).
Please also cite the following papers:
Wei, X., Meel, M.H., Breur, M. et al. Defining tumor-associated vascular heterogeneity in pediatric high-grade and diffuse midline gliomas. acta neuropathol commun 9, 142 (2021). https://doi.org/10.1186/s40478-021-01243-1
Authors’ response: Despite the high scientific interest of the above reference regarding the tumor vasculature in pediatric glioma, in our work we mainly focus on GBM in adult patients. In view of the significant differences between pediatric and adult cancers, we considered that this citation is not fully relevant for this manuscript. We thank the reviewer for sharing this article that might help for further work.
Please see the manuscript in attachment.

Reviewer 3 Report
This is a well-designed study to access the cytotoxicity of Temozolomide and Fingolimod treatments in glioblastoma preclinical models. The novelty of the study resides mostly in the Fingolimod lack of effect in causing cell toxicity in the vitro and animal models used in this study as compared to previous studies showing significant toxicity effects in other peripheral (not CNS) cancers.
The authors also refer the demonstrated growing evidence of sphingosine-1-phosphate (S1P) a lipid-related molecule contributing to cancer and potentially to GBM and that Fingolimod is described as inhibiting S1PR4, as well as S1PR1, S1PR3, and S1PR5. This drug has been shown to retain T cells in secondary lymph organs through the inhibition of S1PRs including S1PR1, S1PR3, S1PR4, and S1PR5 [22,23], limiting T cell brain infiltration and lowering disease progression.
My main concern and question is whether the authors looked for the expression of S1PRs in the cell lines (2D and 3D cultures) used in this study and also in the “tumor microenvironment” of the xenograft model. The results of those experiments could clarify the lack of effect of Fingolimod observed in the present study.
Best regards
Author Response
This is a well-designed study to access the cytotoxicity of Temozolomide and Fingolimod treatments in glioblastoma preclinical models. The novelty of the study resides mostly in the Fingolimod lack of effect in causing cell toxicity in the vitro and animal models used in this study as compared to previous studies showing significant toxicity effects in other peripheral (not CNS) cancers.
The authors also refer the demonstrated growing evidence of sphingosine-1-phosphate (S1P) a lipid-related molecule contributing to cancer and potentially to GBM and that Fingolimod is described as inhibiting S1PR4, as well as S1PR1, S1PR3, and S1PR5. This drug has been shown to retain T cells in secondary lymph organs through the inhibition of S1PRs including S1PR1, S1PR3, S1PR4, and S1PR5 [22,23], limiting T cell brain infiltration and lowering disease progression.
My main concern and question is whether the authors looked for the expression of S1PRs in the cell lines (2D and 3D cultures) used in this study and also in the “tumor microenvironment” of the xenograft model. The results of those experiments could clarify the lack of effect of Fingolimod observed in the present study.
Best regards
Authors’ response: We would like to thank first the reviewer for this very constructive comment on our manuscript and we hope that the response and new edit of the manuscript will convince the reviewer of the interest of our work.
Authors’ response: We would like to thank the reviewer for pointing out this missing information from the original manuscript. Indeed, we demonstrated that U118MG and U87MG are sensitive to Fingolimod in 2D culture but not in other models. We also added 2D culture experiment with GL261 cells upon Fingolimod demonstrated 2D anti-tumor effect (New figure S6 panel C), this information was relevant and was missing from our manuscript. Thus, our cell lines are sensitive to Fingolimod suggesting a potential effect through S1P receptors. Moreover, U87MG cells are described to express all the five S1P receptors at different levels and in particular in orthotopic tumor microenvironment (Annabi et al., Mol Carcinog. 2009 Oct;48(10):910-9. doi: 10.1002/mc.20541. / Bernhart et al., Biochem Pharmacol. 2015 Jul 15; 96(2): 119–130. / Bien-Möller et al., Oncotarget. 2016 Mar 15; 7(11): 13031–13046. / Bien-Möller et al., Cancers 2023, 15(17), 4273). GL261 cells are also described to express S1PR1 and the other S1P receptors at different levels at least in vitro (Bien-Möller et al., Cancers 2023, 15(17), 4273 / Y Xiao · 2018, https://epub.ub.uni-greifswald.de/frontdoor/deliver/index/docId/3956/file/Doctoral+Thesis_yxiao-1.pdf / Marx et al., Cancers (Basel). 2019 Apr; 11(4): 569.). Nevertheless, we cannot exclude that Fingolimod-dependent mechanism observed in 2D culture are S1P receptors-related. Indeed, receptor-independent activity of Fingolimod has been identified via for example the induction of autophagy, the inhibition of histone deacetylases, or binding of protein complex such as inhibitor 2 of protein phosphatase 2A (Wang et al., Curr Med Chem. 2020 Jun; 27(18): 2979–2993.).
The information regarding S1PR expression has now been added to the new version of the manuscript (line 384).
Please see the manuscript in attachment.

Round 2
Reviewer 1 Report
No further comments
Reviewer 3 Report
The major questions have been addressed and the manuscript discussion has been improved.
Best regrads